# Immune-Mediated Bidirectional Causality Between Inflammatory Bowel Disease and Chronic Periodontitis: Evidence from Mendelian Randomization and Integrative Bioinformatics Analysis

**DOI:** 10.3390/biomedicines13020476

**Published:** 2025-02-15

**Authors:** Zhijun Feng, Zihan Chen, Xiaoxu Wang, Meijuan Zhou, Shupeng Liu

**Affiliations:** Department of Radiation Medicine, Guangdong Provincial Key Laboratory of Tropical Disease Research, School of Public Health, Southern Medical University, Guangzhou 510515, China; fengzhj18@lzu.edu.cn (Z.F.); 13288258020@163.com (Z.C.); w15937701495@163.com (X.W.)

**Keywords:** causal relationship, inflammatory bowel disease, periodontitis, Mendelian randomization, integrated bioinformatics analysis

## Abstract

**Background/Objectives**: A bidirectional association between inflammatory bowel disease (IBD) and periodontitis has been observed, yet their causal relationship remains unclear. This study aimed to investigate the potential causal links between these two inflammatory conditions through comprehensive genetic and molecular analyses. **Methods**: We conducted a bidirectional Mendelian randomization (MR) analysis integrated with bioinformatics approaches. The causal relationships were primarily evaluated using inverse variance weighting (IVW), complemented by multiple sensitivity analyses to assess the robustness of the findings. Additionally, we performed differential gene expression analysis using RNA sequencing data to identify co-expressed genes and shared inflammatory mediators between IBD and periodontitis, followed by pathway enrichment analysis. **Results**: Bidirectional MR analysis revealed significant causal associations between IBD and periodontitis (*p*-value < 0.05). Sensitivity analyses demonstrated the consistency of these findings, with no evidence of significant heterogeneity or horizontal pleiotropy (*p*-value > 0.05). Integrated bioinformatics analysis identified key immune regulators, particularly interleukin 1 beta (*IL1B*) and C-X-C motif chemokine receptor 4 (*CXCR4*), and inflammatory signaling pathways, including tumor necrosis factor (*TNF*-α) and interleukin 17 (*IL17*), as potential molecular mechanisms underlying the bidirectional relationship between these conditions. **Conclusions**: Our findings provide genetic evidence supporting a bidirectional causal relationship between IBD and periodontitis. Transcriptomic analysis revealed shared pathological mechanisms and identified crucial immune regulatory factors common to both diseases. These insights enhance our understanding of the molecular interplay between IBD and periodontitis, potentially informing new therapeutic strategies for both conditions.

## 1. Introduction

Periodontal disease is a highly prevalent chronic inflammatory disorder of the oral cavity, with approximately 1.1 billion prevalent cases of severe periodontitis globally in 2019, and has been implicated in numerous systemic pathologies [1,2]. The pathogenesis of this condition is fundamentally linked to perturbations in the oral microbiome composition, wherein pathogenic organisms trigger host immune responses, culminating in tissue inflammation [3]. During this pathological process, a complex network of inflammatory mediators, including cytokines, chemokines, and adenylate cyclase, orchestrates the inflammatory cascade. These molecular mediators not only regulate local inflammatory responses but can also exert systemic effects through hematogenous dissemination. For example, factors like tumor necrosis factor (TNF-α) and interleukin 1 beta (IL1B) not only promote the recruitment and activation of inflammatory cells but may also trigger immune responses in distant tissues, potentially underlying the connection between periodontal disease and systemic illnesses like cardiovascular diseases [4], diabetes [5], and other diseases related to systemic immunity [6]. Moreover, periodontal pathogens and their associated toxins trigger a cascade of inflammatory responses, engaging both innate and adaptive immune mechanisms [7,8]. The resultant inflammatory mediators and immune cells can disseminate systemically through the circulation, extending beyond the confines of the oral cavity to influence distant organ systems [9]. This systemic inflammatory burden, initiated by periodontal disease, may serve as a mechanistic link to various systemic pathologies through sustained immune dysregulation and chronic inflammation.

Inflammatory bowel disease (IBD), including Crohn’s disease (CD) and ulcerative colitis (UC), is another chronic inflammatory condition that impacts gastrointestinal health worldwide, with a global age-standardized prevalence rate of 84.3 per 100,000 population in 2017 [10,11]. These disorders are characterized by persistent inflammation of the intestinal mucosa, resulting in significant gastrointestinal dysfunction. Analogous to periodontal disease, the pathogenesis of IBD involves aberrant immune system activation [12], with pro-inflammatory mediators, including TNF-α, IL-1β, and IL-6, orchestrating the immune microenvironment and driving disease progression [13,14,15]. Furthermore, alterations in gut microbiota-mediated metabolism may significantly influence IBD pathophysiology [16,17], potentially contributing to malignant transformation [18]. Given the shared features of dysregulated inflammatory and immune responses in both periodontal disease and IBD, we have reason to believe that there are underlying pathophysiological connections between these conditions [19,20,21,22]. Beyond the role of inflammatory mediators [23], periodontal pathogens and their associated toxins may directly modulate intestinal inflammation through systemic circulation [24,25]. Moreover, periodontal disease-induced alterations in the oral microbiome may precipitate changes in gut microbiota composition and functionality [26], representing potential mechanistic links between these disorders. Although these conditions differ in their anatomical localization and clinical manifestations, the similarities in their inflammatory and immune response mechanisms provide a theoretical framework for their potential bidirectional interaction. Multiple meta-analyses based on observational studies have confirmed a significant association between periodontitis and IBD [27,28,29,30,31]. However, several recent Mendelian randomization (MR) analyses have reported inconsistent findings [32,33,34,35], making the precise causal relationship between these diseases still elusive. Therefore, further research is needed to clarify the underlying mechanisms linking periodontal disease and IBD, which remains crucial for understanding the precise causal relationship between these conditions.

MR analysis uses genetic variants as instrumental variables (IVs) to reduce the confounding effects common in observational studies, thereby facilitating more robust causal inferences [36,37]. We employed a dual approach to elucidate the relationship between IBD and periodontitis. First, we performed a bidirectional MR analysis to assess potential causal associations based on genetic variation. Second, we conducted transcriptomic analyses of RNA sequencing data from both conditions to identify shared differentially expressed genes (DEGs). Through systematic analysis of gene interaction networks and biological pathway enrichment, we unveiled potential molecular mechanisms underlying the relationship between these conditions. This integrative approach provides novel insights into the molecular interplay between IBD and periodontitis.

## 2. Materials and Methods

### 2.1. Study Design and Data Source

This study incorporated parallel MR and bioinformatics analyses. The MR analysis was conducted in accordance with the Strengthening the Reporting of Mendelian Randomization Studies (STROBE-MR) guidelines [38]. For genome-wide association study (GWAS) data acquisition, we downloaded IBD data [39] (GWAS number: ieu-a-31, Ncase/Ncontrol: 12,882/21,770) and its major subtypes, CD (GWAS number: ieu-a-30, Ncase/Ncontrol: 5956/14,927) and UC (GWAS number: ieu-a-32, Ncase/Ncontrol: 6968/20,464), from the OpenGWAS database (https://gwas.mrcieu.ac.uk/, accessed on 15 October 2024). Chronic periodontitis data (GWAS number: finn-b-K11_PERIODON_CHRON, Ncase/Ncontrol: 3046/195,395) were obtained from the Finngen database (https://www.finngen.fi/, accessed on 15 October 2024) [40]. All datasets comprised large-scale cohorts from distinct geographical regions within the same ethnic population, minimizing potential sample overlap bias.

We performed bidirectional MR analysis. The forward analysis evaluated IBD, CD, and UC as exposure variables with chronic periodontitis as the outcome, while the reverse analysis assessed chronic periodontitis as the exposure with IBD, CD, and UC as outcomes. Single nucleotide polymorphisms (SNPs) associated with exposure variables were selected as IVs, adhering to three fundamental MR assumptions [36,41]: (1) robust association between IVs and exposure variables; (2) independence of IVs from potential confounding factors affecting both exposure and outcomes; and (3) exclusive influence of IVs on outcomes through exposure pathways. As this study utilized publicly available GWAS summary statistics, additional ethical approval and informed consent were not required. Figure 1 shows the analytical workflow, including GWAS details, sample data, SNP metrics, IV selection, and IV counts.

The bioinformatics analysis encompassed systematic investigation of biological processes and molecular regulatory pathways associated with shared differential genes between chronic periodontitis and IBD. This analysis comprised differential gene expression analysis followed by biological process and pathway enrichment analyses. This integrative approach elucidated the immunological mechanisms linking IBD and chronic periodontitis while identifying molecular interactions and pathway convergence points.

### 2.2. Data Cleaning for MR Analysis

The data preprocessing pipeline comprised several sequential steps: Initially, SNPs associated with exposure variables were extracted using the “TwoSampleMR” R package (version 0.6.6) [42,43]. Disease-specific *p*-value thresholds were implemented, 5 × 10^−8^ for IBD-related analyses and 5 × 10^−6^ for periodontitis, accounting for variations in sample size and SNP availability. Subsequently, we eliminated SNPs exhibiting linkage disequilibrium (LD) using stringent parameters (r^2^ = 0.001, window size = 10,000 kb) [44,45]. Instrument strength was assessed through F-statistics, calculated as (Beta^2^/SE^2^) [46], with a minimum threshold of 10 [47,48]. To ensure IV validity, we utilized the LDLink database (https://ldlink.nih.gov/?tab=ldtrait, accessed on 15 October 2024) [49] to identify and exclude potential pleiotropic variants associated with outcome variables in both analytical directions. This systematic approach ensured adherence to core MR assumptions [50]. Identical preprocessing protocols were applied in both forward and reverse analyses to maintain methodological consistency and result comparability.

### 2.3. MR Analysis

The primary MR analysis was conducted using the inverse variance-weighted (IVW)–radial method [51,52], complemented by multiple analytical approaches including IVW with the model of random effects (MRE), IVW with the model of fixed effects (FE), MR-Egger, and weighted median (WM) estimators. To validate the robustness of our findings, we implemented comprehensive sensitivity analyses. Specifically, MR-Egger regression was employed to detect and adjust for potential pleiotropy through intercept testing [53], while the MR-PRESSO method was utilized for outlier detection [54,55]. We conducted leave-one-out analyses to evaluate the influence of individual SNPs on the overall causal estimates [45,56]. Heterogeneity among genetic instruments was assessed using Cochrane’s Q test [37]. All analyses were performed using two-sided tests in R software (version 4.3.3) with the “TwoSampleMR” (version 0.6.6) and “MR-PRESSO” packages (version 1.0). Statistical significance was defined as *p*-value < 0.05 [56].

### 2.4. Obtaining RNA Sequencing Data and Identifying Differential Expressed Genes (DEGs)

The GEO database (http://www.ncbi.nlm.nih.gov/gds, accessed on 20 September 2024) was used for retrieving the RNA sequencing data with the data type set to “Expression profiling by array”, the keywords set to “inflammatory bowel disease” and “periodontitis”, and the species set to “Homo sapiens” [57]. Finally, sequencing data from intestinal mucosa of IBD patients (GSE59071 [58]) and periodontitis patients (GSE16434 [59]) were included in the analysis. Then, using the “limma” R package(version 3.54.2) [60,61] independently, we compared each of these disease-related transcriptomic datasets to their healthy control groups in order to find DEGs. A volcano plot and heatmap will be presented as a display of these differential analyses [62]. The criteria for selecting DEGs were logFC > 1 and *p*-value < 0.01 [63].

### 2.5. Enrichment Analysis and Interaction Network Construction of Co-Expressed DEGs

A visual Venn diagram was drawn and the co-expressed DEGs were extracted for enrichment analysis and interaction network construction. We performed Gene Ontology (GO) and Kyoto Encyclopedia of Genes and Genomes (KEGG) enrichment analyses utilizing the “clusterProfiler” package (version 4.6.2) [64]. The GO analysis was conducted across three categories: biological process (BP), Cellular Component (CC), and Molecular Function (MF) [65]. Subsequently, we applied a significance threshold of adjusted *p*-value < 0.05 to filter the results of both GO and KEGG analyses [66]. The visualization of the outcomes included the presentation of the top 5 GO terms with the most significant *p*-values and the top 10 pathways in the KEGG pathway enrichment analysis with the most significant *p*-values. Finally, the “STRING” database (https://string-db.org/, accessed on 25 October 2024) was utilized to conduct protein–protein interaction (PPI) analysis on the identified co-expressed DEGs [67]. Using “Cytoscape” software (version 3.10.2), we extracted signal strengths and interaction patterns among these co-expressed DEGs by focusing on “Degree” [68].

## 3. Results

### 3.1. The Casual Relationship Between IBD and Chronic Periodontitis

In the forward analysis, we identified 55, 48, and 34 IVs associated with IBD, CD, and UC, respectively (Appendix A). The reverse analysis yielded 48 IVs associated with chronic periodontitis (Appendix A). Following confounder elimination and outcome matching, all identified IVs were incorporated into the bidirectional MR analyses (Appendix A). The bidirectional MR analyses revealed significant causal associations through both IVW–radial and IVW-MRE methods between IBD and chronic periodontitis, particularly for UC (Figure 2, OR > 1, P_IVW-radial_ < 0.05). However, no significant causal relationship was detected between CD and chronic periodontitis in the forward analysis (Figure 2, OR > 1, P_IVW-radial_ > 0.05). In the reverse analysis, all five methods (IVW–radial, IVW-MRE, IVW-FE, MR-Egger, and WM) consistently demonstrated a significant positive causal effect of chronic periodontitis on CD (Figure 2, OR > 1, *p*-value < 0.05). While IVW–radial and IVW-MRE methods indicated a significant causal relationship between chronic periodontitis and IBD (Figure 2, OR > 1, P_IVW-radial_ < 0.05), no significant association was observed between chronic periodontitis and UC (Figure 2, OR > 1, P_IVW-radial_ > 0.05).

### 3.2. Sensitivity Analysis

The estimates of causal effects for each SNP are inconsistent (Figure 3), with some SNPs exhibiting significantly different estimated values than the majority of SNPs or overall MR analysis results, such as rs8178977 (Figure 3A, CD on CP), rs1886731 (Figure 3B, IBD on CP, Figure 3C, UC on CP), rs73404204 (Figure 3D, CP on IDB), rs9932005, and rs73404204 (Figure 3F, CP on UC). Scatter plots of the causal effect estimates of individual SNPs on outcomes for each analysis direction suggested potential outlier distributions (Figure 4). In the leave-one-out analysis test, we found that when removing individual SNPs and repeating the MR analysis, substantial differences were observed in the estimated causal effects (Figure 5), for example, rs10800314 and rs35620072 (Figure 5B, IBD on CP), rs9977672 (Figure 5C, UC on CP), rs115399664 (Figure 5D, CP on CD), rs1986942, rs72894911, rs62567180, and rs78626083 (Figure 5E, CP on IBD); when these SNPs are removed, there will be a significant statistically significant change in the overall causal effect estimation. Notably, despite the fact that there is potential heterogeneity in this analysis from the perspective of a single SNP, no definitive outliers were detected after assessing for outliers using MR-PRESSO (Appendix A). Furthermore, evaluations of heterogeneity using Cochrane’s Q test (Appendix A) and horizontal pleiotropy using the MR-Egger method (Appendix A) did not reveal results with significant statistical evidence (*p*-value > 0.05), indicating that the IVs included in the current bidirectional MR analysis were homogeneous and exhibited no pleiotropy, and all the MR results are robust and reliable.

### 3.3. DEGs in Patients with IBD or Periodontitis

We extracted RNA sequencing data for IBD and periodontitis patients from the GEO database, with each dataset including samples from healthy controls. Differential analysis between the disease cohorts and the healthy controls revealed significant gene expression alterations in patients with periodontitis (Figure 6A,C) and IBD (see Figure 6B,D), predominantly involving genes associated with the immune and inflammatory responses, such as immunoglobulin lambda joining 3 (IGLJ3), C-X-C motif chemokine receptor 4 (CXCR4), CD74 molecule (CD74), complement C2 (C2), among others. These alterations suggest that changes in the expression levels of genes related to immune and inflammatory responses may play a pivotal role in the pathogenesis of periodontitis and IBD, affecting immune cell activity, cytokine release, and immune regulation. This could be a key factor exacerbating the progression of these diseases, implying the potential of these genes as targets for future therapeutic interventions.

### 3.4. Enrichment Analysis and PPI Network of Co-DEGs

In the periodontitis group, a total of 208 DEGs were identified (Appendix A), while in the IBD group, 793 DEGs were determined (Appendix A). From these, we screened out 65 co-expressed DEGs (Figure 7A). GO analysis revealed that these co-expressed DEGs are mainly located in secretory organelles (GO term: CC), and their MF largely involves the regulation of cytokine activity. Biologically, they are primarily associated with the migration of immune cells (GO term: BP, Figure 7B). Further enrichment analysis of the KEGG signaling pathways confirmed (Figure 7C) that these co-expressed DEGs are significantly enriched in cytokine receptor interactions, interleukin-17 (IL-17), and TNF signaling pathways. The molecular interaction analysis of these DEGs revealed an extensive network of inflammation-related factors centered around IL1B and CXCR4 (Figure 7D). This evidence suggests that these molecular interactions could be key in regulating inflammatory responses and pathological processes. They may represent a common pathological characteristic of periodontitis and IBD, potentially providing a basis for a common therapeutic target for these diseases.

## 4. Discussion

The intricate relationship between IBD and chronic periodontitis represents a critical focus in contemporary medical research. Through bidirectional two-sample MR analysis, our study has provided valuable evidence for the mechanistic interplay between these conditions while offering novel therapeutic insights. The MR analysis demonstrated a significant bidirectional causal relationship, indicating that individuals with IBD exhibit increased susceptibility to chronic periodontitis and, conversely, suggesting shared pathogenic mechanisms. To elucidate the molecular basis of this bidirectional relationship, we conducted transcriptomic analyses, which revealed common differentially expressed genes between these conditions. This molecular-level evidence illuminated shared regulatory networks underlying the pathogenesis of both diseases. Subsequent bioinformatics analyses highlighted the central role of immune-related signaling cascades, particularly the IL-17, cytokine, and TNF pathways, in disease progression. Notably, protein–protein interaction analyses identified IL1B and CXCR4 as key immune modulators potentially orchestrating the bidirectional relationship between these conditions. Based on these findings, therapeutic strategies targeting these immune pathways—especially IL-17, TNF, and IL1B—could hold promise for the development of treatments that simultaneously address both IBD and chronic periodontitis, offering a more integrated approach to managing these co-occurring diseases. These findings not only substantiate the hypothesized interaction between IBD and chronic periodontitis but also present novel therapeutic targets. The identification of shared molecular mechanisms and pathological processes offers unprecedented opportunities for developing targeted interventions. This enhanced understanding of the molecular landscape governing both conditions may facilitate more effective, pathway-specific therapeutic strategies, ultimately improving patient outcomes.

IL1B and CXCR4 are key regulators of inflammation and immune responses [69,70], garnering widespread attention for their roles in diseases like IBD and chronic periodontitis in recent years. IL1B is a potent pro-inflammatory cytokine that activates various immune cells and induces the production of other cytokines [71,72]. In IBD, increased expression of IL1B is closely associated with intestinal mucosal inflammation, stimulating intestinal epithelial cells and macrophages to produce inflammatory mediators like prostaglandins and leukotrienes, thereby exacerbating intestinal inflammation [73,74,75]. Additionally, IL1B can promote the infiltration of inflammatory cells, aggravating tissue damage [76]. Similarly, in periodontitis, IL1B plays a crucial role, and it considered one of the primary inflammatory mediators in the disease’s pathogenesis [77,78]. It promotes inflammatory responses in gingival tissues, leading to tissue destruction and bone resorption. CXCR4 is significant in cell migration and immune responses, especially within the inflammatory environments of IBD and chronic periodontitis [79,80,81]. Increased expression of CXCR4 and its ligand CXCL12 in IBD patients aids in the recruitment and activation of inflammatory cells [82,83]. CXCR4 is also involved in maintaining the intestinal epithelial barrier, with its aberrant expression linked to impaired barrier function [84]. In periodontitis, CXCR4 regulates the migration and activation of inflammatory cells in periodontal tissues, influencing disease progression and the extent of tissue damage [83,85].

Apart from IL1B and CXCR4, various other immune-related cytokines play significant roles in the onset and development of IBD and periodontitis. These cytokines regulate inflammation processes through related signaling pathways, such as TNF-α and the IL17 signaling pathway, impacting the clinical manifestations and treatment outcomes of these diseases. TNF-α, a key pro-inflammatory factor, not only enhances the activation of inflammatory cells but also promotes the production and release of cytokines, exacerbating tissue damage [86,87]. The level of TNF is significantly elevated in IBD, correlating with the severity of intestinal inflammation [88]. TNF inhibitors have become an important part of IBD treatment [89,90]. Similarly, in periodontitis, TNF is involved in regulating inflammatory responses and tissue destruction, becoming a potential treatment target [91,92]. IL-17, produced by Th17 cells, is crucial for maintaining the integrity of mucosal barriers [93,94]. In IBD, IL-17 helps enhance mucosal defense, but excessive IL-17 response can exacerbate inflammation [95,96]. In periodontitis, increased IL-17 is also associated with ongoing inflammation and tissue damage [97,98]. Corroborating our findings, recent empirical evidence has demonstrated that periodontal pathogens can exacerbate colitis through modulation of the gut microbiota–metabolite–Th17/Treg axis, further substantiating the mechanistic link between periodontal disease and IBD pathogenesis through immune-mediated pathways [99]. The complex interaction network of these cytokines forms the immunopathological foundation of IBD and periodontitis. Additionally, other interleukin family members, such as interleukin-6 (IL-6)-related pathways, have been reported to be closely associated with inflammation responses in IBD and periodontitis [100,101,102,103]. Understanding the mechanisms of these factors in diseases not only helps to delve deeper into the pathology of IBD and chronic periodontitis but also provides potential targets for developing new treatment methods.

Our findings have several important clinical implications. First, the established bidirectional causal relationship between IBD and chronic periodontitis suggests that clinicians should implement comprehensive screening protocols for both conditions when either is diagnosed, enabling early intervention and improved patient outcomes. Second, the identification of shared molecular pathways, particularly IL-17 and TNF signaling cascades [104,105,106,107], suggests that targeting these pathways could offer a potential avenue for dual-action treatments. Although dual-target therapies have been explored in other contexts, such as in autoimmune diseases like rheumatoid arthritis and psoriasis, their application in managing both IBD and chronic periodontitis remains an area for further investigation. Third, the revelation of key immune regulators, including IL1B and CXCR4 [108], provides new opportunities for developing biomarker-based diagnostic approaches and personalized therapeutic strategies tailored to individual patient profiles.

In comparison to previous studies [19,20,28,31,109,110,111], while several reports have explored the association between IBD and periodontitis, our study provided valuable evidence of a bidirectional causal relationship through the use of MR analysis. Previous research has mostly been observational, often showing correlation rather than causality, and has been limited by confounding factors. Moreover, many studies, including several previously published MR analyses [32,33,34,35], have focused on either the periodontal or gastrointestinal aspects, without investigating the underlying shared molecular mechanisms. By addressing this gap, our research provides a clearer understanding of the interplay between IBD and periodontitis, with a focus on common immune-related signaling pathways. This distinction highlights the need for further investigation into integrated therapeutic strategies targeting these shared mechanisms, which could offer more effective treatment approaches for patients suffering from both conditions simultaneously.

While this study provides valuable insights, several limitations should be acknowledged. First, the MR analysis predominantly focused on European populations, potentially limiting the generalizability of our findings across different ethnic groups due to variations in genetic backgrounds and lifestyle factors [112]. Second, given the systemic nature of IBD, our analysis may not have captured all relevant genetic variations involved in disease pathogenesis, potentially affecting the comprehensiveness of our findings. Third, despite rigorous methodological controls, residual confounding factors may persist, warranting cautious interpretation of the results [113]. Future studies should encompass diverse populations and more extensive genetic analyses to enhance the robustness and applicability of these findings.

## 5. Conclusions

Through MR analysis, this study established a bidirectional causal relationship between IBD and periodontitis, demonstrating increased susceptibility to chronic periodontitis among IBD patients and, conversely, heightened IBD risk in individuals with periodontal disease. Transcriptomic analyses revealed shared differentially expressed genes between these conditions, illuminating common pathogenic mechanisms. Notably, key immune regulators, particularly IL1B and CXCR4, emerged as central mediators in the pathogenesis of both diseases. Furthermore, our findings highlighted the crucial involvement of specific cytokine signaling cascades, especially the IL-17 and TNF pathways, in disease progression. These molecular insights provide a robust scientific framework for the diagnosis and therapeutic management of both conditions, emphasizing the clinical importance of considering their bidirectional relationship in treatment strategies.

## Figures and Tables

**Figure 1 biomedicines-13-00476-f001:**
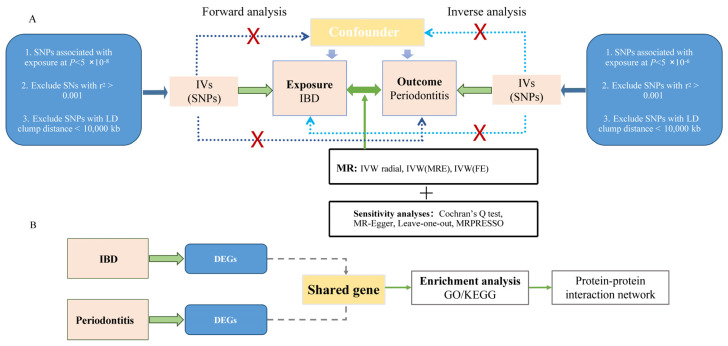
Schematic overview of the study design and analytical workflow. (**A**) Pipeline for Mendelian randomization (MR) analysis demonstrating the systematic evaluation of bidirectional causal relationships between inflammatory bowel disease (IBD) and chronic periodontitis. (**B**) Bioinformatics analysis framework illustrating the integrated transcriptomic and enrichment analyses. LD, linkage disequilibrium; SNP, single nucleotide polymorphism; IVW, inverse-variance weighted; MRE, model of random effect; FE, model of fixed effect; DEGs, differentially expressed genes; KEGG, Kyoto Encyclopedia of Genes and Genomes; GO, gene ontology.

**Figure 2 biomedicines-13-00476-f002:**
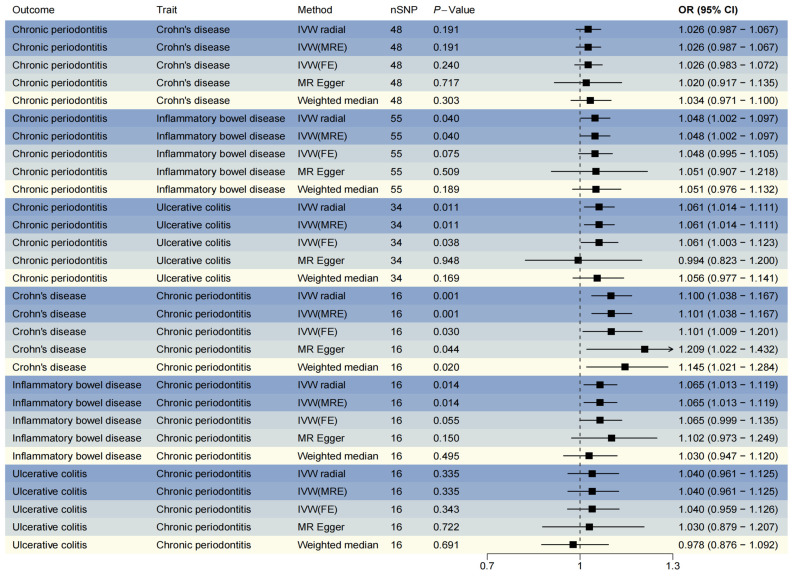
Forest plot for Mendelian randomization (MR) results. IVW, inverse variance-weighted (IVW); MRE, model of random effects; FE, model of fixed effects; nSNP, number of single nucleotide polymorphism; OR, odds ratio; CI, confidence interval. The gradient blue hues symbolize the diverse magnetic resonance (MR) methodologies employed across various analytical dimensions.

**Figure 3 biomedicines-13-00476-f003:**
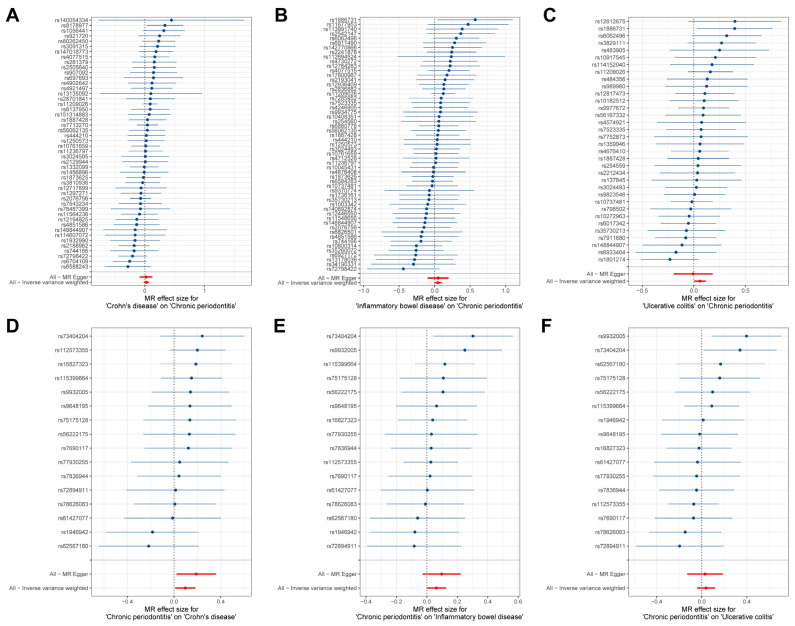
Forest plot for the causal effects estimates of each SNP on outcomes. (**A**–**F**) represent each SNP results for different analytical directions, respectively (with the title description at the bottom of each figure). The blue horizontal line represents the distribution range of the estimated effect value for current SNP, and the blue dots represent the size of the estimated effect value. The vertical line at coordinate 0 is the reference line. The red dots signify the overall causal effect based on MR Egger and inverse variance weighted methods, with the red lines passing through these dots delineating the range of variation in the causal effect.

**Figure 4 biomedicines-13-00476-f004:**
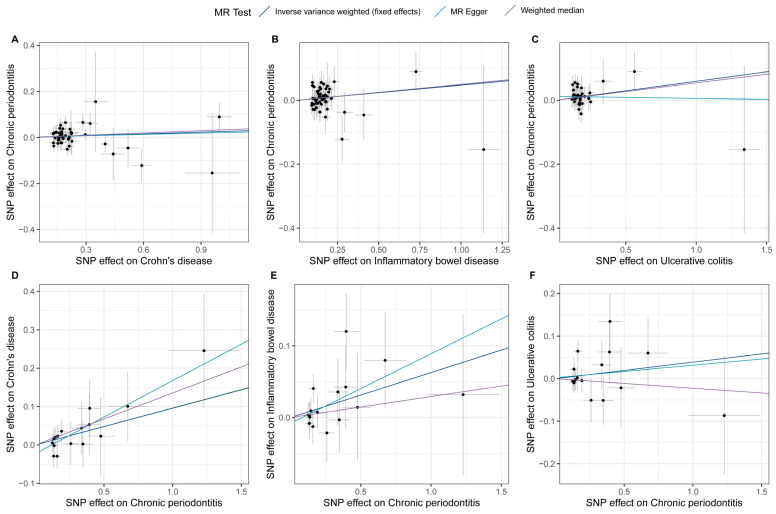
Scatter plots for effects of genetic liability to exposure on outcomes in this bidirectional MR analysis. A to F represent different analytical directions, respectively. Specifically, A-C denote the analyses from Crohn’s disease (**A**), inflammatory bowel disease (**B**), and ulcerative colitis (**C**) to periodontitis; while D-F represent the analyses from periodontitis to Crohn’s disease (**D**), inflammatory bowel disease (**E**), and ulcerative colitis (**F**). Each point represents an included SNP, and the color of the lines corresponds to the methods described in the legend.

**Figure 5 biomedicines-13-00476-f005:**
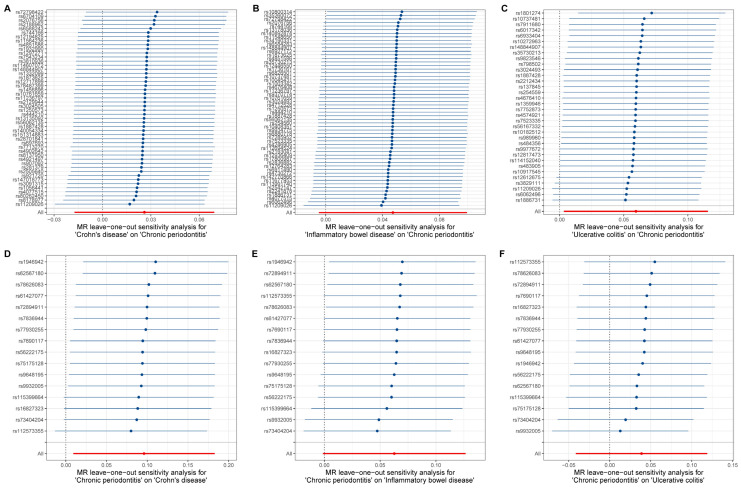
Leave-one-out tests were conducted to evaluate the effects of excluding individual SNPs on the results of MR analysis. (**A**–**F**) represent the leave-one-out analysis results for different analytical directions, respectively (with the title description at the bottom of each figure). The horizontal blue line represents the distribution range of the estimated effect value after removing current single nucleotide polymorphism, and the blue dots represent the size of the estimated effect value. The vertical line at coordinate 0 is the reference line. The red dots signify the overall causal effect, with the red lines passing through these dots delineating the range of variation in the causal effect.

**Figure 6 biomedicines-13-00476-f006:**
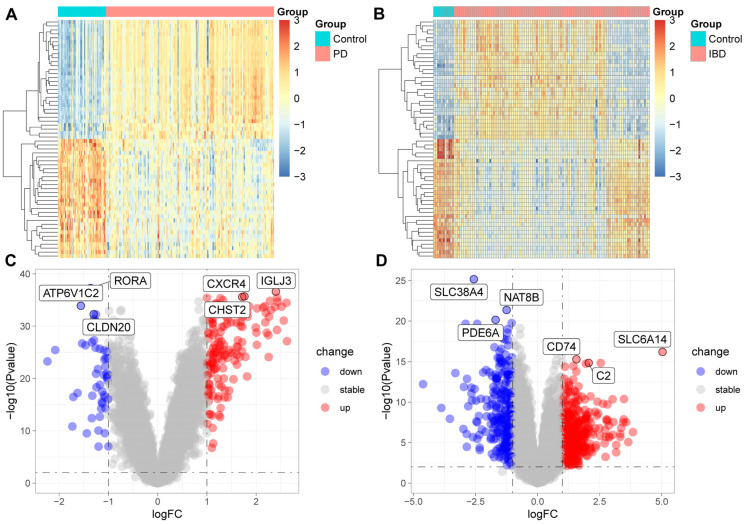
Differential gene expression in patients with periodontitis (**A**,**C**) and IBD (**B**,**D**). PD, chronic periodontitis; down, down-regulated genes, up, up-regulated genes; stable, genes whose expression has no significance between disease and healthy tissues.

**Figure 7 biomedicines-13-00476-f007:**
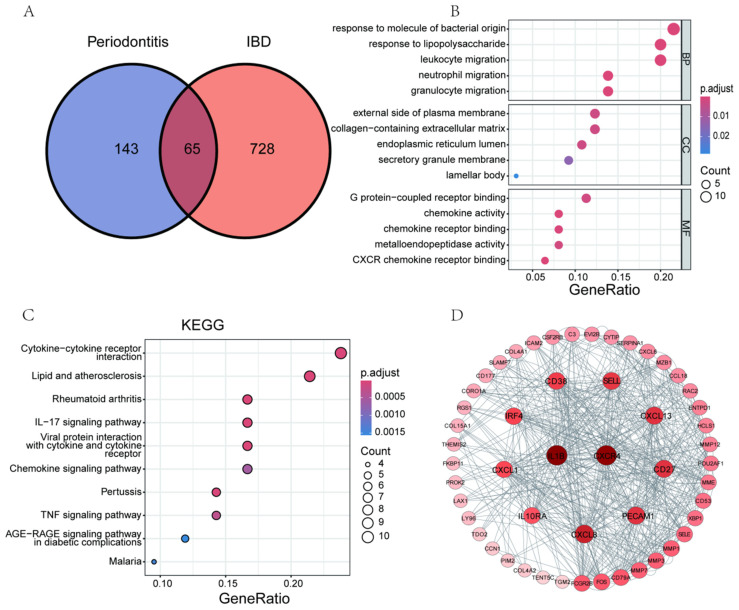
The results of enrichment analysis and protein-protein interaction (PPI) network of Co-DEGs. (**A**) displays a Venn diagram illustrating the distribution of shared and unique differentially expressed genes between periodontitis and inflammatory bowel disease (IBD). (**B**–**D**) present the Gene Ontology(GO) analysis (**B**), KEGG pathway enrichment analysis (**C**), and PPI network results (**D**) for the 65 shared genes between periodontitis and IBD.

## Data Availability

The datasets generated and/or analyzed during the current study are available in the ieu open GWAS project (https://gwas.mrcieu.ac.uk/, accessed on 15 October 2024) and the GEO database (https://www.ncbi.nlm.nih.gov/gds, accessed on 20 September 2024).

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
