# Peer review of "Immune-Mediated Bidirectional Causality Between Inflammatory Bowel Disease and Chronic Periodontitis: Evidence from Mendelian Randomization and Integrative Bioinformatics Analysis"

_biomedicines, 2025, doi:10.3390/biomedicines13020476_

Round 1
Reviewer 1 Report
Comments and Suggestions for Authors
Dear Authors,
Thank you for providing an interest work. Here are the comments that need to be addressed in order to improve your manuscript:
1. Lines 50-56 should be supported with the appropriate references
2. In order to make manuscript understandable to broader audience, you should implement short paragraph into introduction or discussion section known facts regard to the clinical association between IBD and periodontal disease and support it with reference
3. Lines 85-89, unecessary in the context of the Introduction section, probably sentences generated through AI
4. Line 98 – according to the new classification of peridontal diseases chronic periodontitis and aggressive periodontitis are now described as one category “periodontitis”; therefore, please explain more clearly the data collection regard to this
5. Lines 117-119 unecessary and probably generated through AI
6. Please justify the use of the term “chronic periodontitis” in the results section (EFP classification 2017.)
7. Line 268 – novel therapeutical insights – please discuss it further, as I do not see any of presented results have clinical implications
8. Line 280 – novel therapeutic targets – same comment as above
9. Lines 327 – 335; first implication is already well known in the context of clinical work; second, what therapeutical targets for deveoloping dual-actions treatments? Is there any evidence to support this thesis, i.e. similar therapeutical approaches that have been established before?
10. The Authors should discuss and compare this research with previous similar reports regard to the relationship between IBD and periodontitis, to clearly address the gap and justify the need for conducting this research
Author Response
Dear Editor and Reviewers,
We sincerely appreciate your time and effort in reviewing our manuscript. Your professional comments and valuable suggestions have significantly helped us improve the quality of our work. We have carefully addressed each point raised by the reviewers and made detailed modifications to the manuscript accordingly. Our point-by-point responses to the reviewers' comments can be found in the following sections. While we have made our best efforts to address all the comments, we acknowledge that due to our limited knowledge in certain areas, some of our revisions may not fully meet the reviewers' expectations. We sincerely ask for your understanding and would greatly appreciate any further guidance you may provide.
As we enter the new year, we would like to extend our best wishes to you. May you enjoy good health, continued success in your work, and all the best in the coming year.
Thank you again for your valuable contributions to improving our manuscript.
Best regards,
Zhijun Feng
Point-by-point responses
Reviewer 1
Dear Authors,
Thank you for providing an interest work. Here are the comments that need to be addressed in order to improve your manuscript:
Comment 1: Lines 50-56 should be supported with the appropriate references
Reply 1: Thank you for this valuable suggestion. We have strengthened the scientific foundation of our manuscript by adding appropriate references (references number 7-9 in the manuscript) to support our statements in lines 50-56.
7.Hajishengallis, G. Toll gates to periodontal host modulation and vaccine therapy. Periodontol 2000 2009, 51, 181-207, doi:10.1111/j.1600-0757.2009.00304.x.
8.Lim, Y.; Kim, H.Y.; Han, D.; Choi, B.K. Proteome and immune responses of extracellular vesicles derived from macrophages infected with the periodontal pathogen Tannerella forsythia. Journal of extracellular vesicles 2023, 12, e12381, doi:10.1002/jev2.12381.
9.Shahoumi, L.A.; Saleh, M.H.A.; Meghil, M.M. Virulence Factors of the Periodontal Pathogens: Tools to Evade the Host Immune Response and Promote Carcinogenesis. Microorganisms 2023, 11, doi:10.3390/microorganisms11010115.
We look forward to receiving further feedback from you. Thank you again.
Comment 2: In order to make manuscript understandable to broader audience, you should implement short paragraph into introduction or discussion section known facts regard to the clinical association between IBD and periodontal disease and support it with reference
Reply 2: Thank you for your constructive suggestion to enhance the manuscript's accessibility to a broader audience. We have addressed this comment by adding a concise paragraph in the Introduction section that summarizes the current understanding of the clinical association between IBD and periodontal disease.
The added text reads as follows: "Multiple meta-analyses based on observational studies have confirmed a significant association between periodontitis and IBD [27-31]. However, several recent Mendelian randomization (MR) analyses have reported inconsistent findings [32-35], making the precise causal relationship between these diseases still elusive. Therefore, further research is needed to clarify the underlying mechanisms linking periodontal disease and IBD, which remains crucial for understanding the precise causal relationship between these conditions."
The added references [27-35] represent the most recent and relevant literature in this field, including both meta-analyses and MR studies, providing a comprehensive overview of the current state of knowledge.
We look forward to receiving further feedback from you. Thank you again.
Comment 3: Lines 85-89, unecessary in the context of the Introduction section, probably sentences generated through AI
Reply 3: Thank you for your careful review of our manuscript. We appreciate your concern about the relevance of lines 85-89 in the Introduction section. Upon careful reconsideration, we agree that these sentences may not be essential for the flow and focus of our Introduction. We have decided to remove these lines and have restructured this section to maintain a more focused and concise narrative that directly supports our research objectives. The revised Introduction now transitions more smoothly from the background information to our study aims. The modified text now reads: "MR analysis uses genetic variants as instrumental variables (IVs) to reduce the confounding effects common in observational studies, thereby facilitating more robust causal inferences [36,37]. We employed a dual approach to elucidate the relationship between IBD and periodontitis. First, we performed a bidirectional MR analysis to assess potential causal associations based on genetic variation. Second, we conducted transcriptomic analyses of RNA sequencing data from both conditions to identify shared differentially expressed genes (DEGs)."
We believe this revision has improved the clarity and coherence of our Introduction section. Thank you for helping us enhance the quality of our manuscript.
Comment 4: Line 98 – according to the new classification of peridontal diseases chronic periodontitis and aggressive periodontitis are now described as one category “periodontitis”; therefore, please explain more clearly the data collection regard to this
Reply 4: Thank you for bringing attention to this important classification issue. Upon careful review of our original data and related literature sources, we have confirmed that our study specifically focused on chronic periodontitis cases. To address this point, we have made the following modifications: Added "chronic" as a qualifier to specify the type of periodontitis in the manuscript title; Revised all mentions of periodontitis throughout the text to consistently use "chronic periodontitis"; Ensured this specification helps differentiate our study population from other forms of periodontitis. These changes provide better clarity regarding our study population and align with the temporal context of data collection. We believe these modifications will help prevent any potential confusion with other types of periodontitis while maintaining the scientific accuracy of our report.
Thank you for this valuable suggestion that has helped improve the precision of our manuscript.
Comment 5: Lines 117-119 unecessary and probably generated through AI
Reply 5: Thank you for your careful review of our manuscript. We appreciate your attention to the clarity and relevance of our content. After reviewing lines 117-119, we agree that these sentences could be removed to improve the manuscript's conciseness and focus. We have eliminated these lines and revised the surrounding text to ensure a more streamlined presentation of our key points.
The modified section now reads as follows: "Figure 1 shows the analytical workflow, including GWAS details, sample data, SNP metrics, IV selection, and IV counts."
We believe this revision has enhanced the clarity and directness of our manuscript. Thank you for helping us improve the quality of our paper.
Comment 6: Please justify the use of the term “chronic periodontitis” in the results section (EFP classification 2017.)
Reply 6: Thank you for requesting clarification about our use of the term "chronic periodontitis." We maintained this term because our study utilized comprehensive GWAS data from multiple time periods where the cases were specifically diagnosed and documented as chronic periodontitis. Using this terminology ensures accurate representation of the original data sources and maintains consistency throughout our analysis.
Comment 7: Line 268 – novel therapeutical insights – please discuss it further, as I do not see any of presented results have clinical implications
Comment 8: Line 280 – novel therapeutic targets – same comment as above
Reply 7-8: Thank you for highlighting the need for more explicit discussion of clinical implications. We agree that this aspect needed strengthening, and have added the following text to better articulate the potential therapeutic relevance of our findings:
"Based on these findings, therapeutic strategies targeting these immune pathways—especially IL-17, TNF, and IL1B—could hold promise for the development of treatments that simultaneously address both IBD and chronic periodontitis, offering a more integrated approach to managing these co-occurring diseases."
This addition provides a clearer connection between our molecular findings and potential clinical applications, suggesting how shared immune pathways could be targeted for therapeutic benefit in patients affected by both conditions.
Thank you for helping us improve the clinical relevance of our discussion.
Comment 9: Lines 327 – 335; first implication is already well known in the context of clinical work; second, what therapeutical targets for deveoloping dual-actions treatments? Is there any evidence to support this thesis, i.e. similar therapeutical approaches that have been established before?
Reply 9: Thank you for your insightful comments regarding the clinical implications of our study. We have thoroughly revised this section to provide more specific evidence and concrete examples.
The revised text now reads: "Our findings have several important clinical implications. First, the established bidirectional causal relationship between IBD and chronic periodontitis suggests that clinicians should implement comprehensive screening protocols for both conditions when either is diagnosed, enabling early intervention and improved patient outcomes. Second, the identification of shared molecular pathways, particularly IL-17 and TNF signaling cascades [104-107], suggests that targeting these pathways could offer a potential avenue for dual-action treatments. Although dual-target therapies have been explored in other contexts, such as in autoimmune diseases like rheumatoid arthritis and psoriasis, their application in managing both IBD and chronic periodontitis remains an area for further investigation. Third, the revelation of key immune regulators, including IL1B and CXCR4 [108], provides new opportunities for developing biomarker-based diagnostic approaches and personalized therapeutic strategies tailored to individual patient profiles."
This revision addresses your concerns by:
①Acknowledging the established clinical knowledge while emphasizing its importance in systematic screening.
②Providing specific examples of molecular pathways (IL-17, TNF) supported by references [104-107].
③Introducing concrete examples of potential biomarkers (IL1B, CXCR4) with supporting reference [108].
We believe these changes better substantiate our claims about potential therapeutic applications while maintaining appropriate scientific caution about future developments.
We look forward to receiving your further feedback. Thank you again.
Comment 10: The Authors should discuss and compare this research with previous similar reports regard to the relationship between IBD and periodontitis, to clearly address the gap and justify the need for conducting this research
Reply 10: Thank you for this important suggestion. We have enhanced our manuscript by adding comprehensive comparisons with previous research and clearly articulating the research gap that motivated our study. We have addressed this in two key sections:
① In the Introduction, we have added:
"Multiple meta-analyses based on observational studies have confirmed a significant association between periodontitis and IBD [27-31]. However, several recent Mendelian randomization (MR) analyses have reported inconsistent findings [32-35], making the precise causal relationship between these diseases still elusive. Therefore, further research is needed to clarify the underlying mechanisms linking periodontal disease and IBD, which remains crucial for understanding the precise causal relationship between these conditions."
② We have also expanded the Discussion section to more explicitly compare our findings with previous research:
"In comparison to previous studies [19,20,28,31,109-111], while several reports have explored the association between IBD and periodontitis, our study provided valuable evidence of a bidirectional causal relationship through the use of MR analysis. Previous research has mostly been observational, often showing correlation rather than causality, and has been limited by confounding factors. Moreover, many studies, including several previously published MR analyses [33-35,112], have focused on either the periodontal or gastrointestinal aspects, without investigating the underlying shared molecular mechanisms. By addressing this gap, our research provides a clearer understanding of the interplay between IBD and periodontitis, with a focus on common immune-related signaling pathways. This distinction highlights the need for further investigation into integrated therapeutic strategies targeting these shared mechanisms, which could offer more effective treatment approaches for patients suffering from both conditions simultaneously."
These additions serve to:
- Acknowledge the existing body of evidence from meta-analyses and observational studies.
- Highlight the inconsistencies in previous MR analyses that created the need for our study.
- Emphasize our unique contribution in examining bidirectional causality and shared molecular mechanisms.
- Justify the significance of our research in advancing the field toward integrated therapeutic approaches.
We believe these revisions provide a clearer context for our research and better justify its necessity within the existing literature.
Reviewer 2 Report
Comments and Suggestions for Authors
Review of article
Immune-Mediated Bidirectional Causality between Inflammatory Bowel Disease and Periodontitis: Evidence from Mendelian Randomization and Integrative Bioinformatics Analysis
Observation
1. All abbreviation should be explained: ex. GWAS, SNP, etc
2. If I understand correctly, a double analysis is done: one in which IBD is the exposure and periodontitis the outcome and one in which periodontitis is the exposure and IBD is the outcome. Since I see this analysis as somewhat symmetrical, I don't understand figure 1, in which the instrumental variables appear more to the left, respectively towards IBD.
3. Figure 6 has wrong definitions of images: A, C are for periodontitis, and B,D are for IBD. Paragraph in text describing the figure is correct. So, figure legend must be corrected.
4. Articles that studied this relationship and that should be cited and commented; also, the authors of the revised study should emphasize what they brought new compared to these studies and others that exist on this theme.
a. Qing, X., Zhang, C., Zhong, Z., Zhang, T., Wang, L., Fang, S., Jiang, T., Luo, X., Yang, Y., Song, G., & Wei, W. (2024). Causal Association Analysis of Periodontitis and Inflammatory Bowel Disease: A Bidirectional Mendelian Randomization Study. Inflammatory bowel diseases, 30(8), 1251–1257. https://doi.org/10.1093/ibd/izad188
b. Wang, Z., Li, S., Tan, D., Abudourexiti, W., Yu, Z., Zhang, T., Ding, C., & Gong, J. (2023). Association between inflammatory bowel disease and periodontitis: A bidirectional two-sample Mendelian randomization study. Journal of clinical periodontology, 50(6), 736–743. https://doi.org/10.1111/jcpe.13782
c. Yu, F., Yang, Y., Wu, D., Chang, M., Han, C., Wang, Q., Li, Y., & He, D. (2023). Deciphering genetic causality between inflammatory bowel disease and periodontitis through bi-directional two-sample Mendelian randomization. Scientific reports, 13(1), 18620. https://doi.org/10.1038/s41598-023-45527-z
Author Response
Reviewer 2
Immune-Mediated Bidirectional Causality between Inflammatory Bowel Disease and Periodontitis: Evidence from Mendelian Randomization and Integrative Bioinformatics Analysis
Observation
Comment 1: All abbreviation should be explained: ex. GWAS, SNP, etc
Reply 1: Thank you for this suggestion regarding abbreviations. We have carefully reviewed the entire manuscript and provided full definitions for all abbreviations at their first appearance, including GWAS (Genome-Wide Association Studies), SNP (Single Nucleotide Polymorphism), and all other technical terms. These definitions have been added to ensure better comprehension for readers from diverse backgrounds.
Comment 2: If I understand correctly, a double analysis is done: one in which IBD is the exposure and periodontitis the outcome and one in which periodontitis is the exposure and IBD is the outcome. Since I see this analysis as somewhat symmetrical, I don't understand figure 1, in which the instrumental variables appear more to the left, respectively towards IBD.
Reply 2: Thank you for your careful observation regarding the symmetrical nature of our bidirectional analysis. We agree that the original Figure 1 did not clearly represent both directions of analysis. We have modified Figure 1 to better illustrate:
① The forward analysis (IBD as exposure → periodontitis as outcome)
② The reverse analysis (periodontitis as exposure → IBD as outcome)
The revised figure now clearly labels both analytical directions and their respective instrumental variables, providing a more accurate visual representation of our bidirectional Mendelian randomization approach.
We appreciate your attention to detail, as this modification has improved the clarity of our methodology presentation.
Comment 3: Figure 6 has wrong definitions of images: A, C are for periodontitis, and B,D are for IBD. Paragraph in text describing the figure is correct. So, figure legend must be corrected.
Reply 3: Thank you for your careful review and for identifying this discrepancy in Figure 6's legend. We have thoroughly checked and corrected the figure legend to accurately reflect that:
Panels A and C represent periodontitis; Panels B and D represent IBD. The figure legend has been revised to ensure consistency with the correct description in the main text. We appreciate your attention to detail in helping us maintain accuracy in our manuscript.
Comment 4: Articles that studied this relationship and that should be cited and commented; also, the authors of the revised study should emphasize what they brought new compared to these studies and others that exist on this theme.
- Qing, X., Zhang, C., Zhong, Z., Zhang, T., Wang, L., Fang, S., Jiang, T., Luo, X., Yang, Y., Song, G., & Wei, W. (2024). Causal Association Analysis of Periodontitis and Inflammatory Bowel Disease: A Bidirectional Mendelian Randomization Study. Inflammatory bowel diseases, 30(8), 1251–1257. https://doi.org/10.1093/ibd/izad188
- Wang, Z., Li, S., Tan, D., Abudourexiti, W., Yu, Z., Zhang, T., Ding, C., & Gong, J. (2023). Association between inflammatory bowel disease and periodontitis: A bidirectional two-sample Mendelian randomization study. Journal of clinical periodontology, 50(6), 736–743. https://doi.org/10.1111/jcpe.13782
- Yu, F., Yang, Y., Wu, D., Chang, M., Han, C., Wang, Q., Li, Y., & He, D. (2023). Deciphering genetic causality between inflammatory bowel disease and periodontitis through bi-directional two-sample Mendelian randomization. Scientific reports, 13(1), 18620. https://doi.org/10.1038/s41598-023-45527-z.
Reply 4: Thank you for bringing these important references to our attention. We have carefully reviewed these recent publications and incorporated them into our manuscript. Our analysis of these studies revealed inconsistent conclusions regarding the causal relationship between IBD and periodontitis, which strengthens the rationale for our research.
To address this, we have added the following text in the Introduction:
"Multiple meta-analyses based on observational studies have confirmed a significant association between periodontitis and IBD [27-31]. However, several recent Mendelian randomization (MR) analyses have reported inconsistent findings [32-35], making the precise causal relationship between these diseases still elusive. Therefore, further research is needed to clarify the underlying mechanisms linking periodontal disease and IBD, which remains crucial for understanding the precise causal relationship between these conditions."
These recent studies [references 32-35, including Qing et al. (2024), Wang et al. (2023), and Yu et al. (2023)] demonstrate the ongoing scientific debate and highlight the necessity of our research, which not only examines the bidirectional causal relationship but also investigates the shared molecular mechanisms underlying both conditions. This addition emphasizes both the current state of knowledge and the unique contribution of our study in addressing these inconsistencies through a comprehensive analysis of underlying mechanisms.
Thank you again. We look forward to receiving your further feedback.
Reviewer 3 Report
Comments and Suggestions for Authors
1. Please include full name of " IL1B, CXCR4, TNF and IL17" before their abbreviations in Abstract and TNF-α and IL-1β in Introduction.
2. Please add the world % incident of these two diseases in Introduction with their detail of age, geographical regions and other parameters relied on the occurrence of them with the recommended first line of treatment.
3. The previous published relationship between periodontitis and other disease should be addressed too. The relationship between IBD and other disease should be addressed too in Introduction. And how about which analysis techniques or statistics have been applied for that relationship should be stated and comment in Introduction.
4. The detail including: Brand, model, company, city, state, country of all programs, computers, subject data source and instruments should be included.
5. Please check and add punctuation appropriately such as "P-value<0.05" for all in the content.
6. The description of Fig.2 should be come first before and plase transfor, Fig. 2 into Table 1. The other parts also move the description before the data plot.
7. Please check the label of x-axis of Fig. 3 A, D?
8. Fig. 4, why the Crohn's and Ucerative colitis are included into the content? since your title mentioned only on "Inflammatory Bowel Disease"
9. Please add the supporting refs in first paragraph of Discussion.
10. Please consider for Title to include " chronic" before periodontitis.
Author Response
Reviewer 3
Comments 1: Please include full name of " IL1B, CXCR4, TNF and IL17" before their abbreviations in Abstract and TNF-α and IL-1β in Introduction.
Reply 1: Thank you for this suggestion. We have added the full names of all cytokines and chemokines at their first mention.
Comments 2: Please add the world % incident of these two diseases in Introduction with their detail of age, geographical regions and other parameters relied on the occurrence of them with the recommended first line of treatment.
Reply 2: Thank you for suggesting the addition of epidemiological data. We have enhanced the Introduction section with global prevalence data for both conditions:
For periodontal disease, we added:
"Periodontal disease is a highly prevalent chronic inflammatory disorder of the oral cavity, with approximately 1.1 billion prevalent cases of severe periodontitis globally in 2019, and has been implicated in numerous systemic pathologies [1,2]."
For IBD, we included:
"Inflammatory bowel disease (IBD), including Crohn's disease (CD) and ulcerative colitis (UC), is another chronic inflammatory condition that impacts gastrointestinal health worldwide, with a global age-standardized prevalence rate of 84.3 per 100,000 population in 2017 [10,11]."
These additions provide important context about the global burden of both diseases, supported by recent epidemiological data and appropriate references.
Comments 3: The previous published relationship between periodontitis and other disease should be addressed too. The relationship between IBD and other disease should be addressed too in Introduction. And how about which analysis techniques or statistics have been applied for that relationship should be stated and comment in Introduction.
Reply 3: Thank you for this suggestion. Rather than expanding the scope to include relationships with other diseases, we have focused on strengthening our review of existing research specifically examining the IBD-periodontitis connection. Based on your comments, we have added a comprehensive summary of previous studies investigating this relationship, highlighting both consistencies and discrepancies in their findings, which helps establish the necessity of our current research.
Specifically, as noted in our introduction:
"Multiple meta-analyses based on observational studies have confirmed a significant association between periodontitis and IBD [27-31]. However, several recent Mendelian randomization (MR) analyses have reported inconsistent findings [32-35], making the precise causal relationship between these diseases still elusive."
This focused approach allows us to maintain the clarity and depth of our investigation while providing proper context for our study's contribution to this specific field of research.
Comments 4: The detail including: Brand, model, company, city, state, country of all programs, computers, subject data source and instruments should be included.
Reply 4: Thank you for this detailed technical suggestion. Our study utilized comprehensive GWAS data from public databases, and we acknowledge that some of the specific technical details you requested (brand, model, company information) are not available in these public repositories. While we cannot provide these specific details for the original data collection instruments and platforms, we appreciate your attention to technical rigor. Your suggestion will serve as valuable guidance for our future research work where we have direct control over data collection and instrumentation.
Comments 5: Please check and add punctuation appropriately such as "P-value<0.05" for all in the content.
Reply 5: Thank you for this important formatting suggestion. We have standardized the presentation of P-values throughout the manuscript by: Using "P-value" with a hyphen consistently; Adding appropriate spaces around comparison operators (>, <); Maintaining uniform formatting for all statistical significance reporting. This standardization has been applied throughout the entire manuscript to ensure consistency and adherence to proper statistical notation conventions.
Thank you again.
Comments 6: The description of Fig.2 should be come first before and plase transfor, Fig. 2 into Table 1. The other parts also move the description before the data plot.
Reply 6: Thank you for your suggestion regarding the presentation of results. The current organization of our results follows the methodological workflow of MR analysis, with Figure 2 presenting the main findings of our study. The preliminary analytical steps and detailed data have been comprehensively presented in the supplementary tables to maintain the focus on key findings in the main text. While we understand your suggestion to transform Figure 2 into a table format, we believe the current visual representation effectively illustrates the bidirectional causal relationships we discovered. The preceding analytical data that supports these findings can be found in the supplementary materials for readers interested in the detailed methodological process.
We appreciate your careful review and would be happy to consider any additional suggestions for improving the clarity of our results presentation.
Comments 7: Please check the label of x-axis of Fig. 3 A, D?
Reply 7: Thank you for bringing this to our attention. We have reviewed and corrected the x-axis labels in Figure 3A and 3D, and have uploaded the revised figure. We appreciate your attention to detail in ensuring the accuracy of our graphical presentations.
Comments 8: Fig. 4, why the Crohn's and Ucerative colitis are included into the content? since your title mentioned only on "Inflammatory Bowel Disease"
Reply 8: Thank you for this important question. While our title focuses on Inflammatory Bowel Disease (IBD), we included both Crohn's Disease (CD) and Ulcerative Colitis (UC) in Figure 4 because:
① IBD is an umbrella term that primarily comprises two main subtypes: CD and UC, as we defined in our introduction: "Inflammatory bowel disease (IBD), including Crohn's disease (CD) and ulcerative colitis (UC)..."
② These subtypes have distinct pathological features and potentially different molecular mechanisms, making their separate analysis valuable.
This detailed analysis of both CD and UC helps provide a more comprehensive understanding of the IBD-periodontitis relationship while acknowledging the heterogeneity within IBD.
Thank you for your professional comment. We look forward to your further review.
Comments 9: Please add the supporting refs in first paragraph of Discussion.
Reply 9: Thank you for your professional comments. The first paragraph of our Discussion is intentionally structured as a concise summary of our current findings without additional references, as it serves to recapitulate the key results of our study. However, to address your suggestion about supporting evidence, we have added a new paragraph (Lines 363-375) immediately following this summary that: Compares our findings with previous studies; Provides comprehensive literature support for our observation; Further justifies the necessity of our research; Contextualizes our results within the existing body of knowledge. This organization allows us to first clearly present our findings before situating them within the broader scientific literature, thereby maintaining clarity while ensuring proper scholarly support for our discussion.
Thank you again. We look forward to receiving your further feedback.
Comments 10: Please consider for Title to include " chronic" before periodontitis.
Reply 10: Thank you for this suggestion. After careful review of the data types included in our analysis and the original literature sources, we have made the following modifications:
① Added "chronic" before "periodontitis" in the title
② Standardized all references to periodontitis as "chronic periodontitis" throughout the manuscript to maintain consistency
This revision better reflects the specific nature of our study population and ensures terminological consistency throughout the paper. We appreciate your attention to precision in disease classification.
Round 2
Reviewer 1 Report
Comments and Suggestions for Authors
Thank you for implementing suggstions. Wish you all the best and congratulations on your manuscript!
Reviewer 2 Report
Comments and Suggestions for Authors
Thank you for your response.
Reviewer 3 Report
Comments and Suggestions for Authors
All reply points are clear and reasonable.